# *Astragalus polysaccharide* (PG2) Ameliorates Cancer Symptom Clusters, as well as Improves Quality of Life in Patients with Metastatic Disease, through Modulation of the Inflammatory Cascade

**DOI:** 10.3390/cancers11081054

**Published:** 2019-07-25

**Authors:** Wen-Chien Huang, Kuang-Tai Kuo, Oluwaseun Adebayo Bamodu, Yen-Kuang Lin, Chun-Hua Wang, Kang-Yun Lee, Liang-Shun Wang, Chi-Tai Yeh, Jo-Ting Tsai

**Affiliations:** 1Department of Medicine, MacKay Medical College, Taipei 110, Taiwan; 2Division of Thoracic Surgery, Department of Surgery, MacKay Memorial Hospital, Taipei 110, Taiwan; 3Division of Thoracic Surgery, Department of Surgery, Shuang Ho Hospital, Taipei Medical University, New Taipei City 235, Taiwan; 4Division of Thoracic Surgery, Department of Surgery, School of Medicine, College of Medicine, Taipei Medical University, Taipei 110, Taiwan; 5Department of Medical Research and Education, Shuang Ho Hospital, Taipei Medical University, New Taipei City 235, Taiwan; 6Division of Hematology/Oncology, Department of Medicine, Shuang Ho Hospital, Taipei Medical University, New Taipei City 235, Taiwan; 7Biostatistics Center, Taipei Medical University, Taipei 110, Taiwan; 8Department of Dermatology, Taipei Tzu Chi Hospital, Buddhist Tzu Chi Medical Foundation, New Taipei City 235, Taiwan; 9School of Medicine, Buddhist Tzu Chi University, Hualien 970, Taiwan; 10Division of Pulmonary Medicine, Department of Internal Medicine, Shuang Ho Hospital, Taipei Medical University, New Taipei City 235, Taiwan; 11Department of Radiation Oncology, Shuang Ho Hospital, Taipei Medical University, New Taipei City 235, Taiwan; 12Department of Radiology, School of Medicine, College of Medicine, Taipei Medical University, Taipei 235, Taiwan

**Keywords:** *Astragalus polysaccharides*, PG2, cytokine, inflammatory cascade, cancer cachexia, quality of life, QoL, fatigue

## Abstract

**Background**: Improving patients’ quality of life (QoL) is a principal objective of all treatment in any clinical setting, including oncology practices. Cancer-associated inflammation is implicated in disease progression and worsening of patients’ QoL. Conventional anticancer therapeutics while selectively eliminating cancerous cells, are evaded by stem cell-like cells, and associated with varying degrees of adverse effects, thus reducing patients’ QoL. This necessitates novel therapeutic approaches with enhanced efficacy, minimal or no treatment-related adverse effects, and improved QoL in patients with cancer, especially those with metastatic/advance stage disease. **Methods**: Sequel to our team’s previous publication, the present study explores probable effects of *Astragalus polysaccharides* (PG2) on cancer-related inflammatory landscape and known determinants of QoL, as well as the probable link between the two to provide mechanistic insight. In an exploratory double blind randomized controlled trial using patients with metastatic disease (*n* = 23), we comparatively evaluated the therapeutic efficacy of high (500 mg) or low (250 mg) dose PG2 administered intravenously (i.v.), with particular focus on its suggested anti-inflammatory function and the probable effect of same on QoL indices at baseline, then at weeks 4 and 8 post-PG2 treatment. **Results**: All 23 patients with metastatic disease treated with either low or high PG2 experienced reduced pain, nausea, vomiting, and fatigue, as well as better appetite and sleep, culminating in improved global QoL. This was most apparent in the high dose group, with significant co-suppression of pro-inflammatory interleukin (IL)-1β, IL-4, IL-6, IL-13, IL-17, monocytes chemotactic protein (MCP)1, granulocyte-macrophage colony-stimulating factor (GM-CSF), vascular endothelial growth factor (VEGF), tumor growth factor (TGF)-β1, interferon (IFN)-γ, and immune suppressors IL-10 and IL-12. Univariate and multivariate analyses revealed that IL-1β, IL-13 and GM-CSF are independent prognosticators of improved QoL. Conclusion: This proof-of-concept study provides premier evidence of functional association between PG2 anti-inflammatory effects and improved QoL in patients with advanced stage cancers, laying the groundwork for future larger cohort blinded controlled trials to establish the efficacy of PG2 as adjuvant anticancer therapy in metastatic or advanced stage clinical settings.

## 1. Introduction

Cancer is a complex pathological entity, which is by definition are plurifactorial in etiology, multisymptomatic in character, and binary in prognosis—poor or good. Patients with advanced stage or metastatic cancer are characterized by multiple symptoms on referral to or presentation at medical facilities [1,2,3]. Consistent with accruing evidence, these symptoms, such as pain, disturbed sleep, loss of appetite, fatigue, anorexia nervosa, nausea, and vomiting, are seldom orphan symptoms, but rather mostly occur in clusters [1,2,3]. These clusters consisting of ≥2 synchronous, often statistically or clinically related symptoms may or may not share the same etiology [4]. The inter-relatedness of these symptoms, their suggested predictability [1,2], and influence on the perceived welfare or well-being of patients with cancer [4] brings these symptom clusters to the fore as vital indicators of response to therapy and determinants of patients’ quality of life (QoL). 

Like cancer itself, the patients’ QoL is conceptually complex, it is by definition a complex, yet subjective coalescence of the functionality of at least four essential human domains, namely physical, social, occupational, and psychological/emotional [5]. These four domains are reflective of the spectrum of evidence-based symptom clusters in advanced or metastatic cancer, namely, cognitive impairment (confusion, agitation, and urinary incontinence), neuropsychological (anxiety, depression, and insomnia), fatigue (anorexia, weight loss, and tiredness), gastrointestinal (nausea and vomiting), pain and debility, the presence of which are wholly or in part influenced by the age, gender, primary cancer site, and Eastern Cooperative Oncology Group (ECOG) performance status of patients [6,7]. 

While the last two decades has been characterized by advances in diagnostic approaches and improved anticancer therapeutic strategies, the synergy between tumor response to therapeutics, progression-free, overall/disease-specific survival, and patients. The perception of wellbeing or QoL has become a vital focus of increased medical attention and research in oncology practices, especially in the context of current existent treatment-related adverse drug reactions or events (ADR/Es), with potential or putative therapeutic benefits being outweighed by treatment-related deleterious impact on patients’ QoL [8,9]. 

Concurring with Professor Lesley Fallowfield [5], the balance between drug efficacy and ADR/Es remains relevant and necessitates the discovery of new molecular oncotargets, as well as, design and/or development of novel therapeutic strategies therapies, and QoL measurements in oncology clinics. The present study was particularly focused on the tetrad—neuropsychological, fatigue, gastrointestinal, and pain in patients with cancer. Besides pain, nonspecific cancer symptoms, such as fatigue, disturbed sleep, loss of appetite, or anxiety, are not routinely monitored in clinical oncology settings and may not be treated as disease- or treatment- specific toxicities in patients with cancer, subsequently forming a function-limiting or debilitating symptom burden implicated in the causation of non-compliance to chemo- and/or radiotherapy, including dose reduction or withdrawal from treatment, thus, undermining the efficacy of any therapeutic intervention [10].

The clustering of cancer symptoms is suggestive of commonality in the underlying developmental mechanisms of the symptom clusters. The synergistic interaction between tumoral immune cells and inflammatory cytokine cascade in the tumor microenvironment culminate in the cancer progression and patients’ phenotypic symptom clusters. The systemic predominance of pro-inflammatory cytokines is increasingly being associated with pain, fatigue, cognitive impairment, therapy resistance, worsening of QoL and poor prognosis in patients with advanced or metastatic cancer [11,12].

*Astragalus polysaccharides* (PG2), processed into a sterile powdery form, is the bioactive component of dried *astragalus membranaceus (Chinese: Huang-Chi)* roots, and elicits a broad spectrum of documented therapeutic effects, including modulation of the immune system. More specifically, PG2 has been shown to induce differentiation of splenic dendritic cells (DCs), expand the CD11c^high^CD45RB^low^ DC pool, increase the Th1/Th2 ratio, and enhance the T cell-mediated immunity in vitro [13]. Concordantly, there are documentations of the immune-regulating effect of polysaccharopeptide through deregulation of MyD88-mediated signaling pathway [14,15]. However, the probable role of PG2 in the amelioration of cancer symptom clusters and consequent improvement of the QoL in patients with metastatic or advanced stage disease is an evolving theme, and we are learning on-the-go. In fact, while in earlier works, our team provided evidence indicating that higher Karnofsky performance status is predictive of reduced cancer-related fatigue in PG2-treated patients with different cancer types [16], and that PG2 enhances the phenotypic polarization of macrophages, functional maturation of dendritic cells, and T cell-mediated immune responses in patients with non-small cell lung carcinoma (unpublished data), the molecular mechanism underlying these observed and reported findings remains unclear and largely unexplored. In the present study, using a pan-cancer cohort, we explore the effect of varied concentration of PG2 on the neuropsychological, fatigue, gastrointestinal, and pain symptom clusters in patients with cancer, and provide evidence that PG2 ameliorates this tetrad of symptom clusters, as well as improves the QoL in patients with metastatic disease, through modulation of the inflammatory cascade.

## 2. Materials and Methods

### 2.1. Drugs

There were 250 and 500 mg vials of PG2 lyophilized powder obtained from PhytoHealth (PhytoHealth Corporation, Taipei, Taiwan), reconstituted in 10 mL normal saline solution, and then diluted in 490 mL saline solution, ex tempore. Rate of intravenously (i.v.) injection was ~200 mL/h. 

### 2.2. Patient Selection and Study Design

A cohort of 23 subjects with advanced stage progressive metastatic cancer randomly selected from our initial multicenter double-blind randomized phase IV study cohort (*n* = 310) [16]. The inclusion and exclusion criteria are as already outlined previously [16]. As already indicated in our previous publication [16], our study cohort was Pan-Cancer with participants drawn from diverse tumor types, including breast, colorectal, head and neck, liver, lung, pancreas, and stomach. Breast cancer, colorectal cancer, and lung cancer were most frequent, accounting for >40% of the study cohort. The study was approved by the Institutional Review Board (IRB) of the Shuang Ho Hospital and MacKay Memorial Hospital, consistent with the recommendations of the declaration of Helsinki for biomedical research and followed standard institutional protocol for human research. Consistent with the American Society of Clinical Oncology (ASCO) guidelines which advocates against the use of chemotherapy in patients with solid tumor who had shown no curative benefit from prior therapy and for whom the ECOG performance status score was ≥3 [17], no enrolled patient in our cohort (*n* = 23), most of whom (91.3%) had prior to the present study requested an halt to chemotherapy because of ‘perceived’ non-effect or objective worsening of their QoL, were receiving any chemotherapy. All enrolled patients provided written informed consent and were randomly assigned to either the high (500 mg q.d.) or low (250 mg q.d.) dose arm. PG2 was administered three times per week per cycle of 4 weeks. The study started in November 2012 and was completed in June 2017 (Clinical trial information: IRB No.: 201205017 and NCT01720550).

### 2.3. Randomization and Blinding

Patients were assigned to treatment arms randomly with equal probability using a validated auto randomization system, as previously described [16]. Each included patient was assigned a study number with specified treatment plan. The randomization list was only available to the sample manufacturer and was only unblinded after completion of the study.

### 2.4. Drug Safety Evaluation

Drug safety was evaluated as previously described [16]. Briefly, the onset of any new PG2-related adverse drug reactions or events (ADR/Es) or exacerbation of existing ADR/Es, were recorded consistent with the Medical Dictionary for Regulatory Activities (MedDRA 9.1). The safety profiles of enrolled subjects included their vital signs and physical examination result. 

### 2.5. Study Endpoints

The primary study endpoint was changes in inflammatory cytokines, including interleukin (IL)-1β, IL-4, IL-6, Monocyte chemoattractant protein-1 (MCP1/CCL-2), IL-10, IL-12, IL-13, IL-17, Granulocyte-macrophage colony-stimulating factor (GM-CSF/CSF-2), Vascular endothelial growth factor (VEGF), Transforming growth factor beta 1 (TGF-β1), and Interferon gamma (IFN-γ). Secondary endpoints were QoL scores using the European Organization for the Research and Treatment of Cancer (EORTC) Quality of Life Questionnaire (11 questions (SS11) of the EORTC QLQ-C30) and brief fatigue inventory (BFI). These endpoints were monitored at outpatient visits using blood sampling and questionnaires before and after PG2 injection.

### 2.6. Determination of Serum Inflammatory Cytokine Levels

The patients’ IL-1β, IL-4, IL-6, MCP-1, IL-10, IL-12, IL-13, IL-17, GM-CSF, VEGF, TGF-β1, and IFNγ sera cytokine level were determined using the Cytokine Human Magnetic 30-Plex Panel for the Luminex^TM^ platform (Cat. No. LHC6003M. Thermo Fisher Scientific Inc., Waltham, MA, USA) and Luminex^®^100/200^TM^ system (Luminex Corporation, Austin, TX, USA). Sera were stored at −80 °C until use and processed strictly following the manufacturer’s instructions. The quantification criteria were based on fluorescence values of 100 events per region. For cytokine standard curve generation in duplicate, we serially diluted the recombinant standards provided in the kit. Curves were adjusted to a five-parameter logistic regression model with correlation coefficients (R^2^) > 0.95, and the sera cytokine levels (pg/mL) were interpolated from the standard curves. 

### 2.7. Statistical Analysis

All data are expressed as mean ± standard deviation (SD), 95% confidence interval (95% CI), and median/interquartile range (IQR), as appropriate. Comparison between two groups of categorical data was analyzed using the Mann-Whitney *U* test or Pearson’s χ2 test, while the significance of differences between time-points was determined using the non-parametric Wilcoxon signed-rank test. For comparison between the two groups of continuous variables, the independent student’s *t-*test was used. EORTC and cytokines were analyzed using the log-rank test, Cox’s proportional hazards univariate and multivariate regression model. All statistical analyses were performed using Statistical Product and Service Solutions, SPSS (IBM Corp. Released 2016. IBM SPSS Statistics for Windows, Version 24.0., IBM, Armonk, NY, USA). *p*-value < 0.05 was considered statistically significant. 

## 3. Results

### 3.1. Patient Characteristics

Of the original 310 subjects screened for eligibility, 23 who apparently met all the inclusion criteria were enrolled for this study and their data analyzed. This present study enrolled 11 from the original 154 in the high dose arm, and 12 from the original 156 in the low dose arm of our PG2 randomized double-blind clinical trial [16]. All 23 participants (100%) completed the baseline assessment as well as both primary and secondary endpoint evaluation. All participants had locally advanced or metastatic cancer and were under palliative treatment at the time of PG2 initiation. The general clinico-demographic characteristics of our study cohort, namely age, gender, body mass index (BMI), brief fatigue inventory (BFI), and Karnofsky performance status (KFS) are summarized in Table 1.

### 3.2. PG2 Elicits Significant Improvement in Patients’ QoL

Evaluation of patients’ QoL using the EORTC QLQ-C30 SS11 showed that all determinants of QoL including pain, sleep, appetite, nausea, and fatigue started to improve even by first visit (visit 1) and were significantly improved by visit 3 after infusion of high or low dose of PG2. However, we find it confounding that while an appreciable decreasing trend was observed in vomiting, with no observed or reported case of emesis on visit 3, it was not statistically significant for both treatment arms (low: *p* = 0.84; high: *p =* 0.363) (Table 2). The observed improvement in QoL is reflects time-dependence and/or pharmaco-cumulative effect of PG2. Interestingly, significant improvement in the global QoL was reported by patients in both low dose (*p =* 0.02) and high dose (*p =* 0.012) arms of PG2 infusion. This post-PG2 improvement in QoL was both time- and dose-dependent (Table 2). With regards to fatigue, the findings of the present study showing that PG2 significantly ameliorates cancer-related fatigue in patients with metastatic or advanced stage disease are consistent at least in part with those of our previous publication [16]. 

### 3.3. PG2 Infusion Down-Modulate Patients’ Pro-Inflammatory Cytokine Profiles

PG2-induced alteration in patients’ sera cytokine profile determined using the Cytokine Human Magnetic 30-Plex Panel for the Luminex^TM^ platform and Luminex^®^100/200^TM^ system revealed that compared to the baseline sera cytokine levels, PG2 induced significant reduction in the production of inflammatory cytokines IL-1β (*p =* 0.014), IL-4 (*p =* 0.04), IL-6 (*p =* 0.012), IL-10 (*p =* 0.019), IL-12 (*p =* 0.03), IL-13 (*p =* 0.025), IL-17 (*p =* 0.018), TGF-β1 (*p =* 0.024), IFN-γ (*p =* 0.031), GM-CSF (*p =* 0.014), VEGF (*p =* 0.023), and chemokine MCP1 (*p =* 0.017) in our metastatic disease cohort. This reduction was apparent as early as at the first visit (visit 1) on week 4 of cycle 1 (c1.4) after PG2 initiation, and was time-dependent, however, a strict dependence on dose was not inferable from available data (Table 3, also see Appendix A). It is worth noting that aside the suppression of pro-inflammatory cytokines (IL-1β, IL-4, IL-6, IL-13, IL-17, IFN-γ and other growth factors (TGF-β1, GM-CSF, and VEGF), and chemokine (MCP1), PG2 also markedly suppressed anti-inflammatory cytokines IL-10 and IL-12 over the course of our study.

### 3.4. PG2-Induced Improvement in QoL Is Associated with Changes in Patients’ Inflammatory Cytokine Profiles

Cox’s proportional hazards univariate and multivariate regression modeling of the EORTC and alteration in patients’ sera cytokine levels indicated a functional association between the inflammatory cytokine levels and patients’ QoL based on the symptom burden. Drawing from the nine symptom subscales of the EORTC QLQ-C30, namely fatigue, nausea/vomiting, pain, dyspnoea, insomnia, appetite loss, constipation, diarrhea, and financial difficulties, as well as the global health/QoL subscale, we established a positive correlation between inflammatory cytokine levels and pain, nausea, vomiting and fatigue, appetite loss and disturbed sleep/insomnia, and poor QoL, as summarized in Table 4, whereby for every unit increase in the predictor variable (cytokine level), the outcome variable (EORTC QLQ-C30) increases by the indicated beta coefficient value. These data are suggestive of negatively correlated PG2-induced improvement in patients’ QoL and changes in the sera level of inflammatory cytokines IL-1β (Pr > |t| = 0.0004), IL-4 (Pr > |t| = 0.0006), IL-6 (Pr > |t| = 0.0002), IL-10 (Pr > |t| = 0.0017), IL-12 (Pr > |t| = 0.0041), IL-13 (Pr > |t| = 0.0013), IL-17 (Pr > |t| = 0.001), IFN-γ (*p =* 0.031) TGF-β1 (Pr > |t| = 0.0011), GM-CSF (Pr > |t| < 0.001), VEGF (Pr > |t| = 0.0009) and chemokine MCP1 (Pr > |t| = 0.0007) are statistically significant as shown by univariate linear regression. Concordantly, our multivariate linear regression analyses revealed that IL-1β (β = 0.2531, Pr > |t| = 0.0013), IL-13 (β = −0.085, Pr > |t| = 0.0442) and GM-CSF (β = 0.1266, Pr > |t| = 0.002) are independent prognosticators of improved QoL (Table 4).

### 3.5. PG2, the Inflammatory Cascade, and the QoL of Patients with Metastatic Cancer

Functional enrichments in the PG2-modulated inflammatory cytokines were analyzed by using the STRING: functional protein association networks (STRINGdb) (https://string-db.org) platform for visualization of protein-protein interaction networks. Consistent with our thinking, with an average local clustering coefficient of 0.964, and protein-protein interaction (PPI) enrichment *p*-value < 1.0 × 10^−16^, our interactome analysis of PG2-modulated IL-1β, IL-4, IL-6, MCP1, IL-10, IL-12, IL-13, IL-17, GM-CSF, VEGF, TGF-β1, and IFN-γ demonstrate that the network formed by these biomarkers exhibited significantly more interactions (*n* = 63) between each other than expected (*n* = 12) for a random set of proteins of similar size, thus indicating the cytokines are at least, in part, biologically connected as a group (Figure 1A). The concept of a functionally connected group is result of our gene co-expression analysis in human and other organisms (Figure 1B). Gene ontology (GO) classification and Kyoto encyclopedia of genes and genomes (KEGG) pathways—based clustering implicated the group in immune response and inflammatory signaling, respectively (data not provided). In addition, hierarchical cluster analysis (HCA) was performed based on the PG2 treatment regimen and changes in our cohort cytokine profile, with the dendrograms defining 2 sample groups (high dose: HC1.1, HC1.4, HC2.1, HC2.4, and low dose: LC1.1, LC1.4, LC2.1, LC2.4), and characterized by a single cluster of our cohort’s inflammatory cytokines, namely IL-1β, IL-4, IL-6, MCP1, IL-10, IL-12, IL-13, IL-17, GM-CSF, VEGF, TGF-β1, and IFN-γ. Our cluster distribution analysis showed changes in the inflammatory cytokines profile was statistically significant for HC1.4 (*p =* 0.013), HC2.1 (*p =* 0.021), HC2.4 (*p <* 0.0001), LC1.4 (*p =* 0.048), LC2.1 (*p =* 0.0053), and LC2.4 (*p* < 0.01), with synchronous reduction in expression of all the inflammatory cytokines was observed in both the high and low dose groups (Figure 1C). 

## 4. Discussion

This translational study provides premier evidence of the functional association between PG2 anti-inflammatory effects and improved QoL in patients with advanced stage cancers. Our study demonstrated that PG2 concomitantly elicits significant amelioration of the symptom burden or clusters and reduction in the expression level and/or activity of major pro-inflammatory cytokines, including IL-1β, IL-6, IL-12, IFN-γ, and GM-CSF, as early as within a the first month after low or high dose PG2 initiation in patients with metastatic or advanced stage cancer. While our study reveals the expedited therapeutic and QoL-enhancing benefits of PG2 in a clinical setting, we are cautious in the interpretation of our findings to avoid over-interpretation of our data, especially considering the relative small cohort size of the present study and its inherent lack of the conventional treatment (untreated or placebo-treated) control group. 

These findings are clinically relevant for medical decision making and selection of therapeutic strategy, especially as cumulative evidence implicates current anticancer therapy (chemotherapy and/or radiation therapy) in occurrence of ADR/Es in patients with cancer undergoing treatment. In fact, though limited, available data does indicate that approximately a tenth of the global population and up to a fifth (20%) of hospitalized patients experience ADR/Es [17,18], such as depression, loss of appetite, fatigue, disturbed sleep, cognitive difficulties, nausea, vomiting, and pain. Regardless of the unitary or clustered nature of these symptoms, they are associated with exacerbated QoL, with diminished physical, psychological, and social functioning. Thus, the present clinical study represents an important progress in providing some mechanistic insight into the occurrence of cancer-related symptom clusters, their inverse association with patients’ QoL, and the inflammation-mediated therapeutic activity of PG2. 

Regardless of its origin, whether causally-associated with anticancer therapeutics, or inherently characteristic of advanced disease, cancer-related symptom clusters bear adversely on the QoL [5,6,7,8,9,10] and are independently associated with the survival of patients with metastatic disease [19,20]. Therefore, the demonstrated ability of PG2 to ameliorate cancer-related symptom clusters, as well as improve the quality of life in patients with metastatic disease, through modulation of the inflammatory cascade targeting is consistent with progressive therapeutic trends in clinical oncology practice. The clinical use of PG2 as adjunct therapy to target inflammation in patients with metastatic cancer is a therapeutic approach with significant positive implications for the alleviation of patients’ symptom burden and enhancement of the clinical efficacy of standard of care anticancer chemotherapy, considering that majority of cancer-related symptom clusters are not only associated with systemic inflammation, but are essentially driven by same, thus, the comeliness of our findings in the context of an highly efficacious symptom cluster-alleviating, QoL-improving, inflammation-limiting, anticancer adjuvant therapy, as proposed by Laird BJ, et al. [21].

Begging its case for the clinical feasibility and applicability of PG2 as an efficacious anticancer adjuvant/adjunct therapy, the administration of PG2, markedly improved our patients’ QoL based on apparent amelioration in seven out of the 15 QoL domains of EORTC QLQ-C30 assessed in this study, namely pain, disturbed sleep, loss of appetite, nausea, vomiting, fatigue, and global QoL, as early as on first visit (visit 1; C1.4), and even more significantly by visit 3 (C2.4). This ameliorative effect of PG2 on metastatic disease-inherent or chemotherapy-induced pain, disturbed sleep, loss of appetite, nausea, vomiting, fatigue, and poor global QoL in our metastatic or advanced stage disease cohort was time-dependent and/or pharmaco-cumulative in nature. This is clinically relevant when taken in the context of recent findings by Trajkovic-Vidakovic, M, et al. [22], showing that patients’ QoL is severely impaired in the weeks following the administration of conventional chemotherapy, as demonstrated by increased symptom burden in nine out of the 15 QoL domains of the EORTC QLQ-C30 as early as a week after chemotherapy administration [23].

Mechanistically, the deregulation of the inflammatory cascade and down-modulation of its associated signals by PG2 with marked amelioration in pain, disturbed sleep, loss of appetite, nausea, vomiting, and fatigue domains of the EORTC QLQ-C30, and resultant QoL improvement is consistent with scientific rationalization and corroborated by another recent work by Laird BJ, et al. [21] on the differential association of QoL in patients with advanced cancer with performance status and systemic inflammatory response. Herein, they demonstrated that increasing modified Glasgow Prognostic Score (mGPS; systemic inflammation) and deteriorating performance status (PS) are associated with the deterioration in QoL parameters (*p <* 0.001), indicating that enhanced systemic inflammation is associated with deterioration in patients’ QoL parameters independent of PS [24]. Thus, we posit that coupled with the immense discomfort, diminished or loss of function, and physical or psychosocial distress inflicted by pain, disturbed sleep, loss of appetite, nausea, emesis, and fatigue on patients with metastatic disease, these adverse determinants of QoL invariably curtail patients’ tolerance to anticancer therapy. It thus becomes incumbent on healthcare providers in oncology practices to ensure that these treatment-related and/or advanced disease-inherent cancer symptom clusters are effectively managed to enhance treatment outcome—both subjective (as perceived by the patients) and objective (as confirmed by physical examination and biochemical and instrumental investigations). Thus, as rightly put by Charles Cleeland, “a mechanistic understanding of treatment-related symptoms would be of benefit in drug development, drug evaluation and early integration of appropriate supportive care in treatment planning” [25]. Herein, lays the benefit of PG2 as adjuvant therapy; to improve the health status of these patients, while concurrently minimizing function-impairing anticancer treatment-related toxicities, increasing therapy compliance, boosting health-related QoL and potentially conferring some survival advantage. 

The resultant effect of PG2 treatment highlights the therapeutic efficacy and medical validity of inflammatory cascade blockade as an effective therapeutic approach for patients with metastatic or advanced stage disease. Combining results of our present study with those from our teams previously published work [16]; PG2 is a potential inflammation-targeting therapeutic option for managing patients with metastatic and advanced disease patients. It is noteworthy that univariate and multivariate analyses suggest a strong positive correlation between EORTC QLQ-C30-based QoL and the responsiveness of the inflammatory cytokines IL-1β, IL-4, IL-6, MCP1, IL-10, IL-12, IL-13, IL-17, GM-CSF, VEGF, TGF-β1, and IFN-γ. Nonetheless, considering possible biases associable with the limitations of the study design which include a histologically-diverse cancer cohort and clinical ethics-based lack of conventional no treatment or placebo control group, we cautiously draw (non-) definitive conclusions based on the significant difference between the baseline and eventual post-PG2 treatment QoL variables of the patients. However, this study lays the foundation for further large cohort prospective randomized clinical studies to clarify if and to what extent the responsiveness of the inflammatory cytokine cascade is a reliable and objective predictor of the QoL for patients with metastatic cancer receiving PG2.

## 5. Conclusions

This clinical study highlights the disease-relevance of changes in systemic levels of inflammatory cytokines, including IL-1β, IL-6, IL-12, GM-CSF, TGF-β1, and IFN-γ as well as the therapeutic efficacy of PG2 as a cancer symptom cluster-alleviating, QoL-improving, inflammation-limiting, adjunct therapy for patients with metastatic or advanced stage cancer. To the best of our knowledge, this is the first study providing evidence that PG2 ameliorates cancer symptom clusters, as well as improves quality of life in patients with metastatic disease, through down-modulation of the inflammatory cascade. 

## Figures and Tables

**Figure 1 cancers-11-01054-f001:**
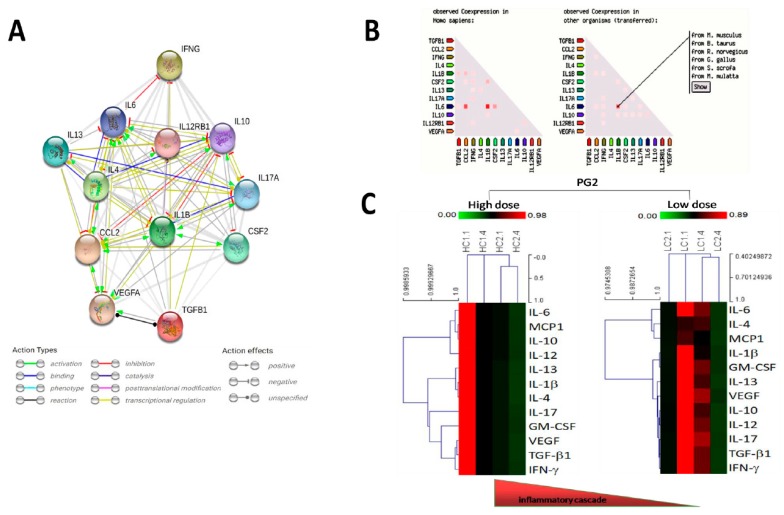
*Astragalus polysaccharides* (PG2)-induced improvement in Quality of life (QOL) is associated with changes in patients’ inflammatory cytokine profiles. (**A**) Visualization of the protein-protein interaction networks of the inflammatory cytokines of our cohort using STRING: functional protein association networks (STRINGdb) platform. (**B**) Representative data from gene coexpression analysis of the inflammatory cytokines of our cohort. (**C**) Unsupervised hierarchical cluster heatmap showing the high or low dose PG2-induced changes in expression of our cohort inflammatory cytokines over the study course.

**Table 1 cancers-11-01054-t001:** Cohort clinico-demographic characteristics.

Variable	Astragalus Polysaccharides (PG2) Dosage	
	High (500 mg)	Low (250 mg)
Gender (*n*,%)		
Male	3 (27.27%)	7 (58.33%)
Female	8 (72.73%)	5 (41.67%)
Age (years)		
Mean (SD)	67.27 (12.31)	63.67 (14.65)
Medial (IQR)	64 (20)	62 (20.5)
95% CI	(59.00–75.54)	(54.36–72.97)
Body mass index (BMI)(kg/m^2^)		
Mean (SD)	21.22 (3.38)	21.08 (4.13)
Medial (IQR)	21.17 (5)	20.67 (5.81)
95% CI	(18.95–23.50)	(18.45–23.70)
Baseline BFI score (Cycle 1 visit 1)		
Mean (SD)	8.37 (1.18)	7.79 (1.67)
Medial (IQR)	8.30 (2.20)	8.25 (2.70)
95% CI	(7.58–9.16)	(6.72–8.87)
Baseline Karnofsky Performance status score (Cycle 1 visit 1)		
Mean (SD)	70.71 (10.44)	65.00 (15.08)
Medial (IQR)	70 (20)	70 (25)
95% CI	(63.89–77.93)	(55.42–74.58)

BFI: brief fatigue inventory; SD: standard deviation; IQR: interquartile range; CI: confidence interval.

**Table 2 cancers-11-01054-t002:** European Organization for the Research and Treatment of Cancer (EORTC) quality of life questionnaire (QLQ)-C30.

Domains	PG2	Baseline	Visit 1	Visit 2	Visit 3	p-value for change
Pain	High	48.48 ± 26.30	26.67 ± 27.89	21.78 ± 22.77	15.00 ± 16.67	0.038
	Low	50.00 ± 36.24	34.44 ± 17.21	23.33 ± 17.89	13.33 ± 33.33	0.014
Sleep	High	78.79 ± 22.47	40.00 ± 43.46	33.33 ± 31.62	8.33 ± 16.67	0.003
	Low	56.94 ± 41.72	44.44 ± 17.21	35.33 ± 34.56	22.22 ± 19.25	0.057
Appetite	High	66.67 ± 29.81	33.33 ± 23.57	30.56 ± 19.48	16.67 ± 19.25	0.013
	Low	68.06 ± 34.42	38.89 ± 25.09	30.00 ± 34.56	22.22 ± 38.49	0.095
Nausea	High	27.27 ± 25.03	13.33 ± 18.26	10.89 ± 19.48	6.89 ± 19.48	0.046
	Low	30.56 ± 28.28	22.22 ± 27.22	16.67 ± 43.46	16.67 ± 43.46	0.093
Vomiting	High	12.12 ± 23.68	7.33 ± 18.26	5.56 ± 13.61	0.00 ± 0	0.363
	Low	11.11 ± 16.41	11.11 ± 17.21	20.00 ± 44.72	0.00 ± 0	0.841
Fatigue	High	86.36 ± 17.98	40.00 ± 14.91	33.33 ± 14.91	8.33 ± 16.67	0.003
	Low	86.11 ± 21.12	33.33 ± 21.08	43.33 ± 32.49	22.22 ± 19.25	0.08
Global QoL	High	30.30 ± 15.49	56.67 ± 25.28	70.83 ± 13.69	83.33 ± 13.69	0.012
	Low	24.31 ± 18.28	61.11 ± 17.21	60.00 ± 25.95	77.78 ± 9.62	0.02
Total score	High	54.82 ± 8.06	39.09 ± 10.62	38.63 ± 8.07	31.06 ± 6.72	0.006
	Low	50.56 ± 9.11	43.68 ± 5.28	44.54 ± 17.78	34.34 ± 6.99	0.004

EORTC: European organization for the research and treatment of cancer; C30: core 30; QoL: quality of life.

**Table 3 cancers-11-01054-t003:** PG2-induced alteration in patients’ inflammatory cytokine profile.

Cytokines	Baseline	Visit 1	Visit 2	Visit 3	p-value for change
IL-1β	155.5 ± 44.3	131.8 ± 40.4	119.02 ± 2.15	104.4 ± 3.8	0.014
IL-4	143.1 ± 28.6	132.4 ± 39.9	120.54 ± 4.3	102.8 ± 2.7	0.04
IL-6	147.4 ± 28	133.1 ± 39.4	122.15 ± 1.16	103.6 ± 3.2	0.012
MCP-1	143.5 ± 28.3	131.2 ± 40.9	121.63 ± 0.42	103.6 ± 3.3	0.017
IL-10	152.6 ± 45.3	132.4 ± 39.9	122.78 ± 2.06	103.6 ± 3.3	0.019
IL-12	151.1 ± 46.4	133 ± 39.5	122.62 ± 1.82	104.1 ± 2.8	0.03
IL-13	154.3 ± 46.1	132.4 ± 39.9	120.00 ± 3.54	102.8 ± 2.8	0.025
IL-17	153.4 ± 44.9	133.7 ± 38.9	120.23 ± 3.87	102.9 ± 2.7	0.018
GM-CSF	160.3 ± 45.8	133 ± 39.5	119.54 ± 2.88	105.2 ± 3	0.014
VEGF	153.7 ± 44.8	133.6 ± 39	120.23 ± 3.87	105.6 ± 3.6	0.023
TGF-β1	153.3 ± 44.9	132.9 ± 39.5	120.17 ± 3.78	105.1 ± 3.2	0.024
IFN-γ	151.3 ± 46	133 ± 39.5	120.17 ± 3.77	104.7 ± 2.7	0.031

IL: interleukin; MCP-1: Monocyte chemoattractant protein-1; GM-CSF: Granulocyte-macrophage colony-stimulating factor; VEGF: Vascular endothelial growth factor; TGF-β1: Transforming growth factor beta 1; IFN-γ: Interferon gamma.

**Table 4 cancers-11-01054-t004:** Linear regression estimates for EORTC QLQ-C30 with selected cytokines.

Cytokines	Univariate		Multivariate	
	β	Pr > |t|	β	Pr > |t|
IL-1β	0.043	0.0004	0.2531	0.0013
IL-4	0.051	0.0006		
IL-6	0.051	0.0002		
MCP-1	0.046	0.0007		
IL-10	0.037	0.0017		
IL-12	0.034	0.0041		
IL-13	0.04	0.0013	−0.085	0.0442
IL-17	0.041	0.001		
GM-CSF	0.047	<0.001	0.1266	0.002
VEGF	0.041	0.0009		
TGF-β1	0.041	0.0011		
IFN-γ	0.038	0.002		

IL: interleukin; MCP-1: Monocyte chemoattractant protein-1; GM-CSF: Granulocyte-macrophage colony-stimulating factor; VEGF: Vascular endothelial growth factor; TGF-β1: Transforming growth factor beta 1; IFN-γ: Interferon gamma.

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
