# Peer review of "Astragalus polysaccharide (PG2) Ameliorates Cancer Symptom Clusters, as well as Improves Quality of Life in Patients with Metastatic Disease, through Modulation of the Inflammatory Cascade"

_cancers, 2019, doi:10.3390/cancers11081054_

Round 1
Reviewer 1 Report
Cancer treatment focused in enhanced efficacy with a minimal or no treatment-related adverse effects, and improved QoL in patients with cancer, especially those with metastatic/advance stage disease is necessary.
This study provides of functional association between PG2 anti inflammatory effects and improved QoL in patients with advanced stage cancers, in metastatic or advanced stage clinical settings.
Author Response
Point-by-point responses to reviewer’s comments:
We would like to thank the reviewers for the thorough reading of our manuscript as well as their valuable comments. We have followed their comments closely and feel that they have further strengthened the manuscript. Below are our point-by-point responses.
Q1: Reviewer #1: Cancer treatment focused in enhanced efficacy with a minimal or no treatment-related adverse effects, and improved QoL in patients with cancer, especially those with metastatic/advance stage disease is necessary.
A1: This study provides of functional association between PG2 anti-inflammatory effects and improved QoL in patients with advanced stage cancers, in metastatic or advanced stage clinical settings. We sincerely thank the reviewer for the time taken to review our work, and for the suggestions given. In this revised manuscript, we have made use of the reviewer’s suggestion.
Q1: Reviewer #2: Here Author checked the improvement of QoL of patients with metastatic disease by modulation of inflammatory cascade. However, i found limitations in the paper.
A1: We sincerely thank the reviewer for the time taken to review our work, and for the comments made. In this revised manuscript (R1), we have made use of the reviewer’s critiques and suggestions, and hope the changes and editions made herein will be satisfactory and help allay the reviewer’s concerns.

Reviewer 2 Report
Here Author checked the improvement of QoL of patients with metastatic disease by modulation of inflammatory cascade. However i found limitations in the paper:
1) Were the patients on any anticancer therapy? Please provide details of each patients.
2) Did you check the effect of PG2 on metastasis progression?
3)Providing graphs of individual cytokine of each patients in supplementary would be informational.
Author Response
Q1: Reviewer #2: Here Author checked the improvement of QoL of patients with metastatic disease by modulation of inflammatory cascade. However, i found limitations in the paper.
A1: We sincerely thank the reviewer for the time taken to review our work, and for the suggestions given. In this revised manuscript, we have made use of the reviewer’s critiques and suggestions, and hope this allay the reviewer’s concerns.
Q2: Reviewer #2: Were the patients on any anticancer therapy? Please provide details of each patients.
A2: We really thank the reviewer for this comment. We believe this is a legitimate question arising from the initial lack of clarity of our manuscript. As now clarified in our revised manuscript, our cohort consisted of 23 subjects with advanced stage progressive metastatic cancer characterized by an ECOG score of 3 or greater and as such were not ‘under active cancer therapies’. Please kindly see our revised Materials and Methods, Pages 3-4, Lines 132-150.
2.2. Patient Selection and Study Design
A cohort of 23 subjects with advanced stage progressive metastatic cancer randomly selected from our initial multicenter double-blind randomized phase IV study cohort (n = 310) (16). The inclusion and exclusion criteria are as already outlined previously (16). As already indicated in our previous publication (16), our study cohort was Pan-Cancer with participants drawn from diverse tumor types, including breast, colorectal, head and neck, liver, lung, pancreas, and stomach. Breast cancer, colorectal cancer, and lung cancer were most frequent, accounting for > 40% of the study cohort. The study was approved by the Institutional Review Board (IRB) of the MacKay Memorial Hospital, consistent with the recommendations of the declaration of Helsinki for biomedical research and followed standard institutional protocol for human research. Consistent with the American Society of Clinical Oncology (ASCO) guidelines which advocates against the use of chemotherapy in patients with solid tumor who had shown no curative benefit from prior therapy and for whom the ECOG performance status score was ≥ 3 (17), no enrolled patient in our cohort (n = 23), most of whom (91.3%) had prior to the present study requested an halt to chemotherapy because of ‘perceived’ non-effect or objective worsening of their QoL, were receiving any chemotherapy. All enrolled patients provided written informed consent and were randomly assigned to either the high (500mg q.d) or low (250mg q.d) dose arm. PG2 was administered three times per week per cycle of 4 weeks. The study started in November 2012 and was completed in June 2017 (Clinical trial information: NCT01720550).
Also kindly see our revised References section, Page 12, Lines 431-433.
17. Prigerson HG, Bao Y, Shah MA, Paulk E, LeBlanc TW, Schneider BJ, et al. Chemotherapy Use, Performance Status, and Quality of Life at the End of Life. JAMA Oncol. 2015; 1(6):778-784. doi:10.1001/jamaoncol.2015.2378
Q3: Reviewer #2: Did you check the effect of PG2 on metastasis progression?
A3: A3: We sincerely appreciate this comment by the reviewer. This present study did not explore the effect of PG2 on metastasis as a biological even per se. It is however the subject of an on-going sub-study. Preliminary results from functional assays including the scratch-wound healing migration, Transwell Matrigel invasion, and colony formation assays, as well as western blot analysis of selected markers of EMT in cells exposed to varying concentrations of PG2, are quite interesting and promising. Hopefully, we will share this with the general public as soon as the sub-study is completed.
Q4: Reviewer #2: Providing graphs of individual cytokine of each patient in supplementary would be informational.
A4: We thank the reviewer for this comment. We have now provided data of the ‘individual cytokine of each patients’ as requested by the reviewer. Please kindly see the Supplementary Data section, Supplementary Figure S1 with legend, Page 13.
3.3. PG2 Infusion Down-Modulate Patients’ Pro-Inflammatory Cytokine Profiles
PG2-induced alteration in patients’ sera cytokine profile determined using the Cytokine Human Magnetic 30-Plex Panel for the LuminexTM platform and Luminex®100/200TM system revealed that compared to the baseline sera cytokine levels, PG2 induced significant reduction in the production of inflammatory cytokines IL-1β (p = 0.014), IL-4 (p = 0.04), IL-6 (p = 0.012), IL-10 (p = 0.019), IL-12 (p = 0.03), IL-13 (p = 0.025), IL-17 (p = 0.018), TGF-β1 (p = 0.024), IFN-γ (p = 0.031), GM-CSF (p = 0.014), VEGF (p = 0.023) and chemokine MCP1 (p = 0.017) in our metastatic disease cohort. This reduction was apparent as early as at the first visit (visit 1) on week 4 of cycle 1 (c1.4) after PG2 initiation, and was time-dependent, however, a strict dependence on dose was not inferable from available data (Table 3, also see Supplementary Table S1 and Supplementary Figure S1). It is worth noting that aside the suppression of pro-inflammatory cytokines (IL-1β, IL-4, IL-6, IL-13, IL-17, IFN-γ and other growth factors (TGF-β1, GM-CSF, VEGF) and chemokine (MCP1), PG2 also markedly suppressed anti-inflammatory cytokines IL-10 and IL-12 over the course of our study.
Supplementary Figure S1. Graphical representation of the individual inflammatory cytokine level in each patient from our advanced progressive metastatic cancer cohort.
Supplementary Table S1. PG2-induced alteration in patients' inflammatory cytokine profile at baseline, first, and last visits.
Baseline | v1 | v3 | ||
IL-1b | High | 154.8±33 | 130.4±15.27 | 103.13±3.19 |
Low | 156.23±54.25 | 132.97±55.44 | 106.1±4.55 | |
combined | 155.5±44.3 | 131.8±40.4 | 104.4±3.8 | |
IL-4 | High | 153.35±33.39 | 130.4±15.27 | 101.31±0.24 |
Low | 133.61±20.44 | 134.07±54.7 | 104.86±3.29 | |
combined | 143.1±28.6 | 132.4±39.9 | 102.8±2.7 | |
IL-6 | High | 155.9±33.17 | 130.4±15.27 | 102.82±3.19 |
Low | 139.54±20.6 | 135.3±53.9 | 104.66±3.64 | |
combined | 147.4±28 | 133.1±39.4 | 103.6±3.2 | |
MCP1 | High | 153.63±33.26 | 130.4±15.27 | 104.2±3.85 |
Low | 134.21±20.03 | 131.95±56.17 | 102.7±2.8 | |
combined | 143.5±28.3 | 131.2±40.9 | 103.6±3.3 | |
IL-10 | High | 153.08±33.54 | 130.4±15.27 | 104.2±3.85 |
Low | 152.25±55.5 | 134.14±54.68 | 102.7±2.8 | |
combined | 152.6±45.3 | 132.4±39.9 | 103.6±3.3 | |
IL-12 | High | 150.81±35.66 | 130.4±15.27 | 104.96±3.03 |
Low | 151.41±56.1 | 135.12±53.99 | 103.02±2.5 | |
combined | 151.1±46.4 | 133±39.5 | 104.1±2.8 | |
IL-13 | High | 151.72±34.63 | 130.4±15.27 | 102.48±3.41 |
Low | 156.75±56.16 | 134.15±54.64 | 103.18±2.34 | |
combined | 154.3±46.1 | 132.4±39.9 | 102.8±2.8 | |
IL-17 | High | 153.08±33.54 | 130.4±15.27 | 101.16±0.45 |
Low | 153.75±54.79 | 136.46±53.07 | 105.13±2.84 | |
combined | 153.4±44.9 | 133.7±38.9 | 102.9±2.7 | |
GM-CSF | High | 157.99±34.56 | 130.4±15.27 | 104.2±3.85 |
Low | 162.5±55.59 | 135.12±53.99 | 106.42±0.96 | |
combined | 160.3±45.8 | 133±39.5 | 105.2±3 | |
VEGF | High | 153.35±33.39 | 130.4±15.27 | 103.79±3.44 |
Low | 154±54.7 | 136.18±53.25 | 108.06±2.49 | |
combined | 153.7±44.8 | 133.6±39 | 105.6±3.6 | |
TGF-b1 | High | 153.08±33.54 | 130.4±15.27 | 105.79±3.48 |
Low | 153.5±54.88 | 135.02±54.07 | 104.1±3.06 | |
combined | 153.3±44.9 | 132.9±39.5 | 105.1±3.2 | |
IFN-g | High | 151.72±34.63 | 130.4±15.27 | 104.96±3.03 |
Low | 151±56.09 | 135.12±53.99 | 104.31±2.7 | |
combined | 151.3±46 | 133±39.5 | 104.7±2.7 |

Reviewer 3 Report
In this manuscript, Huang et al reported findings from a clinical study in which advanced stage cancer patients were treated with two doses of Astragalus polysaccharides (PG2). The data indicated that PG2 treatment improved patient’s quality of life (QoL) through EORTC assessments. Accompanying the improved QoL, the authors also found significant reduction in a panel of pro-inflammatory/anti-inflammatory cytokines, and derived a cluster of affected cytokine network based on the cytokine expression data. While the topic of this study demonstrates novelty, more in-depth analysis of the biology and mechanisms is encouraged. Below are a few suggestions for improvement:
1. For the patient cohort, were all patients diagnosed with the same type of cancer? If it was population with mixed cancer types, does different cancer types affect the clinical outcome of PG2 treatment?
2. Were the patients enrolled in this study under active cancer therapies? Could the effect of improved QoL or decreased cytokines be a consequence of other therapies instead of PG2?
3. Does Table 3 combine the cytokine results from low dose and high dose groups? If so, please split apart because they were two separate arms of the clinical study.
4. The authors noticed a significant drop in large panel of cytokines post PG2 treatment, which might suggest a severe disruption in patient’s immune system. They should provide more insights into the patient’s immune populations such as T cell, B cell, neutrophil, monocytes, DC/macrophages. They should also discuss more on whether the sharp drop of cytokines would sensitize patients to other complications.
Author Response
Q1: Reviewer #3: In this manuscript, Huang et al reported findings from a clinical study in which advanced stage cancer patients were treated with two doses of Astragalus polysaccharides (PG2). The data indicated that PG2 treatment improved patient’s quality of life (QoL) through EORTC assessments. Accompanying the improved QoL, the authors also found significant reduction in a panel of pro-inflammatory/anti-inflammatory cytokines and derived a cluster of affected cytokine network based on the cytokine expression data. While the topic of this study demonstrates novelty, more in-depth analysis of the biology and mechanisms is encouraged. Below are a few suggestions for improvement
A1: We sincerely thank the reviewer for the time taken to review our work, and for the comments made. In this revised manuscript (R1), we have made use of the reviewer’s critiques and suggestions, and hope the changes and editions made herein will be satisfactory and help allay the reviewer’s concerns.
Q2: Reviewer #3: For the patient cohort, were all patients diagnosed with the same type of cancer? If it was population with mixed cancer types, does different cancer types affect the clinical outcome of PG2 treatment?
A2: We sincerely thank the reviewer for this insightful comment. As indicated in our previous publication (16), the patient cohort consisted of patients diagnosed with different types of cancer. We have now indicated this is the revised manuscript. Please kindly see our revised Materials and Methods section, Pages 3 – 4, Lines 132 – 150.
2.2. Patient Selection and Study Design
A cohort of 23 subjects with advanced stage progressive metastatic cancer randomly selected from our initial multicenter double-blind randomized phase IV study cohort (n = 310) (16). The inclusion and exclusion criteria are as already outlined previously (16). As already indicated in our previous publication (16), our study cohort was Pan-Cancer with participants drawn from diverse tumor types, including breast, colorectal, head and neck, liver, lung, pancreas, and stomach. Breast cancer, colorectal cancer, and lung cancer were most frequent, accounting for > 40% of the study cohort. The study was approved by the Institutional Review Board (IRB) of the MacKay Memorial Hospital, consistent with the recommendations of the declaration of Helsinki for biomedical research and followed standard institutional protocol for human research. Consistent with the American Society of Clinical Oncology (ASCO) guidelines which advocates against the use of chemotherapy in patients with solid tumor who had shown no curative benefit from prior therapy and for whom the ECOG performance status score was ≥ 3 (17), no enrolled patient in our cohort (n = 23), most of whom (91.3%) had prior to the present study requested an halt to chemotherapy because of ‘perceived’ non-effect or objective worsening of their QoL, were receiving any chemotherapy. All enrolled patients provided written informed consent and were randomly assigned to either the high (500mg q.d) or low (250mg q.d) dose arm. PG2 was administered three times per week per cycle of 4 weeks. The study started in November 2012 and was completed in June 2017 (Clinical trial information: NCT01720550).
Also kindly see our Reference section, Page 12, Lines 427 – 429.
16. Wang CH, Lin CY, Chen JS, Ho CL, Rau KM, Tsai JT, Chang CS, Yeh SP, Cheng CF, Lai YL. Karnofsky Performance Status as A Predictive Factor for Cancer-Related Fatigue Treatment with Astragalus Polysaccharides (PG2) Injection-A Double Blind, Multi-Center, Randomized Phase IV Study. Cancers (Basel). 2019 Jan 22; 11(2). pii: E128.
Regarding the second part of the reviewer’s question, the results and findings of the present study were regardless of the different cancer types constituting the study cohort, as we observed that the effect of PG2 on clinical outcome and QoL were not affected by the histological origin of the cancer types represented. We have now indicated this in the revised manuscript. Please kindly see our revised Conclusions section, Page 11, Lines 369 – 376.
5. Conclusions
This clinical study highlights the disease-relevance of changes in systemic levels of inflammatory cytokines, including IL-1β, IL-6, IL-12, GM-CSF, TGF-β1, and IFN-γ as well as the therapeutic efficacy of PG2 as a cancer symptom cluster-alleviating, QoL-improving, inflammation-limiting, adjunct therapy for patients with metastatic or advanced stage cancer, regardless of histological origin of the cancers in question. To the best of our knowledge, this is the first study providing evidence that PG2 ameliorates cancer symptom clusters, as well as improves quality of life in patients with metastatic disease, through down-modulation of the inflammatory cascade.
Q3: Reviewer #3: Were the patients enrolled in this study under active cancer therapies? Could the effect of improved QoL or decreased cytokines be a consequence of other therapies instead of PG2?
A3: We really thank the reviewer for this comment. We believe this is a legitimate question arising from the initial lack of clarity of our manuscript. As now clarified in our revised manuscript, our cohort consisted of 23 subjects with advanced stage progressive metastatic cancer characterized by an ECOG score of 3 or greater and as such were not ‘under active cancer therapies’. Please kindly see our revised Materials and Methods, Pages 3-4, Lines 132-150.
2.2. Patient Selection and Study Design
A cohort of 23 subjects with advanced stage progressive metastatic cancer randomly selected from our initial multicenter double-blind randomized phase IV study cohort (n = 310) (16). The inclusion and exclusion criteria are as already outlined previously (16). As already indicated in our previous publication (16), our study cohort was Pan-Cancer with participants drawn from diverse tumor types, including breast, colorectal, head and neck, liver, lung, pancreas, and stomach. Breast cancer, colorectal cancer, and lung cancer were most frequent, accounting for > 40% of the study cohort. The study was approved by the Institutional Review Board (IRB) of the MacKay Memorial Hospital, consistent with the recommendations of the declaration of Helsinki for biomedical research and followed standard institutional protocol for human research. Consistent with the American Society of Clinical Oncology (ASCO) guidelines which advocates against the use of chemotherapy in patients with solid tumor who had shown no curative benefit from prior therapy and for whom the ECOG performance status score was ≥ 3 (17), no enrolled patient in our cohort (n = 23), most of whom (91.3%) had prior to the present study requested an halt to chemotherapy because of ‘perceived’ non-effect or objective worsening of their QoL, were receiving any chemotherapy. All enrolled patients provided written informed consent and were randomly assigned to either the high (500mg q.d) or low (250mg q.d) dose arm. PG2 was administered three times per week per cycle of 4 weeks. The study started in November 2012 and was completed in June 2017 (Clinical trial information: NCT01720550).
Also kindly see our revised References section, Page 12, Lines 431-433.
17. Prigerson HG, Bao Y, Shah MA, Paulk E, LeBlanc TW, Schneider BJ, et al. Chemotherapy Use, Performance Status, and Quality of Life at the End of Life. JAMA Oncol. 2015; 1(6):778-784. doi:10.1001/jamaoncol.2015.2378
Q4: Reviewer #3: Does Table 3 combine the cytokine results from low dose and high dose groups? If so, please split apart because they were two separate arms of the clinical study.
A4: We thank the reviewer for this insightful comment. Yes, as indicated in the manuscript Table 3 combines the serum cytokine profile data from low and high dose groups, since as already stated in the text; no dependence on PG2 dose could be inferred during the study. As requested by the reviewer’s and to allay any concerns, we have now provided a dose-based segregated data as supplementary data in our revised manuscript. Please kindly see our revised Results section, Page 6, Lines 217-229.
Supplementary Table S1. PG2-induced alteration in patients' inflammatory cytokine profile at baseline, first, and last visits.
Baseline | v1 | v3 | ||
IL-1b | High | 154.8±33 | 130.4±15.27 | 103.13±3.19 |
Low | 156.23±54.25 | 132.97±55.44 | 106.1±4.55 | |
combined | 155.5±44.3 | 131.8±40.4 | 104.4±3.8 | |
IL-4 | High | 153.35±33.39 | 130.4±15.27 | 101.31±0.24 |
Low | 133.61±20.44 | 134.07±54.7 | 104.86±3.29 | |
combined | 143.1±28.6 | 132.4±39.9 | 102.8±2.7 | |
IL-6 | High | 155.9±33.17 | 130.4±15.27 | 102.82±3.19 |
Low | 139.54±20.6 | 135.3±53.9 | 104.66±3.64 | |
combined | 147.4±28 | 133.1±39.4 | 103.6±3.2 | |
MCP1 | High | 153.63±33.26 | 130.4±15.27 | 104.2±3.85 |
Low | 134.21±20.03 | 131.95±56.17 | 102.7±2.8 | |
combined | 143.5±28.3 | 131.2±40.9 | 103.6±3.3 | |
IL-10 | High | 153.08±33.54 | 130.4±15.27 | 104.2±3.85 |
Low | 152.25±55.5 | 134.14±54.68 | 102.7±2.8 | |
combined | 152.6±45.3 | 132.4±39.9 | 103.6±3.3 | |
IL-12 | High | 150.81±35.66 | 130.4±15.27 | 104.96±3.03 |
Low | 151.41±56.1 | 135.12±53.99 | 103.02±2.5 | |
combined | 151.1±46.4 | 133±39.5 | 104.1±2.8 | |
IL-13 | High | 151.72±34.63 | 130.4±15.27 | 102.48±3.41 |
Low | 156.75±56.16 | 134.15±54.64 | 103.18±2.34 | |
combined | 154.3±46.1 | 132.4±39.9 | 102.8±2.8 | |
IL-17 | High | 153.08±33.54 | 130.4±15.27 | 101.16±0.45 |
Low | 153.75±54.79 | 136.46±53.07 | 105.13±2.84 | |
combined | 153.4±44.9 | 133.7±38.9 | 102.9±2.7 | |
GM-CSF | High | 157.99±34.56 | 130.4±15.27 | 104.2±3.85 |
Low | 162.5±55.59 | 135.12±53.99 | 106.42±0.96 | |
combined | 160.3±45.8 | 133±39.5 | 105.2±3 | |
VEGF | High | 153.35±33.39 | 130.4±15.27 | 103.79±3.44 |
Low | 154±54.7 | 136.18±53.25 | 108.06±2.49 | |
combined | 153.7±44.8 | 133.6±39 | 105.6±3.6 | |
TGF-b1 | High | 153.08±33.54 | 130.4±15.27 | 105.79±3.48 |
Low | 153.5±54.88 | 135.02±54.07 | 104.1±3.06 | |
combined | 153.3±44.9 | 132.9±39.5 | 105.1±3.2 | |
IFN-g | High | 151.72±34.63 | 130.4±15.27 | 104.96±3.03 |
Low | 151±56.09 | 135.12±53.99 | 104.31±2.7 | |
combined | 151.3±46 | 133±39.5 | 104.7±2.7 |
Q5: Reviewer #3: The authors noticed a significant drop in large panel of cytokines post PG2 treatment, which might suggest a severe disruption in patient’s immune system. They should provide more insights into the patient’s immune populations such as T cell, B cell, neutrophil, monocytes, DC/macrophages. They should also discuss more on whether the sharp drop of cytokines would sensitize patients to other complications.
A5: We really appreciate the reviewer’s insightful comment. Indeed, noting that PG2 induced significant reduction in the production of inflammatory cytokines, we had same concerns and did actually probe for any detrimental immunological dysfunction. While we found none based on clinical, biochemical and instrumental investigations, we unraveled a correlation between the suppressed inflammatory cascade and cellular components of the patients’ immune system, especially the monocyte-macrophage system and this is the subject of another paper which is currently in revision with another journal, and thus, the findings cannot be divulged herei
